# Combination of Total-Reflection X-Ray Fluorescence Method and Chemometric Techniques for Provenance Study of Archaeological Ceramics

**DOI:** 10.3390/molecules28031099

**Published:** 2023-01-21

**Authors:** Artem S. Maltsev, Nailya N. Umarova, Galina V. Pashkova, Maria M. Mukhamedova, Dmitriy L. Shergin, Vitaly V. Panchuk, Dmitry O. Kirsanov, Elena I. Demonterova

**Affiliations:** 1Institute of the Earth’s Crust, Siberian Branch of the Russian Academy of Sciences, 128 Lermontov St., 664033 Irkutsk, Russia; 2Institute of Petroleum, Chemistry and Nanotechnologies, Kazan National Research Technological University, 68 Karl Marx St., 420015 Kazan, Russia; 3Pedagogical Institute, Irkutsk State University, 1 Karl Marx St., 664003 Irkutsk, Russia; 4Institute of Chemistry, St. Petersburg University, 7-9 Universitetskaya Embankment, 199034 St. Petersburg, Russia

**Keywords:** TXRF, archaeological ceramics, PCA, SVM, k-means cluster analysis, provenance, eastern Siberia

## Abstract

The provenance study of archaeological materials is an important step in understanding the cultural and economic life of ancient human communities. One of the most popular approaches in provenance studies is to obtain the chemical composition of material and process it with chemometric methods. In this paper, we describe a combination of the total-reflection X-ray fluorescence (TXRF) method and chemometric techniques (PCA, k-means cluster analysis, and SVM) to study Neolithic ceramic samples from eastern Siberia (Baikal region). A database of ceramic samples was created and included 10 elements/indicators for classification by geographical origin and ornamentation type. This study shows that PCA cannot be used as the primary method for provenance purposes, but can show some patterns in the data. SVM and k-means cluster analysis classified most of the ceramic samples by archaeological site and type with high accuracy. The application of chemometric techniques also showed the similarity of some samples found at sites located close to each other. A database created and processed by SVM or k-means cluster analysis methods can be supplemented with new samples and automatically classified.

## 1. Introduction

The study of samples of ancient ceramics is an important source of information about the cultural and economic life of ancient human communities. This is because ceramics were objects of everyday use and may provide a lot of information about trade relations, religious customs, and communication between various communities [1]. The provenance analysis of ceramics is possible through the determination of chemical composition [2]. Quantitative data regarding elemental composition are usually obtained by analyzing powdered ceramic fragments through X-ray fluorescence spectrometry (XRF) [3,4,5], instrumental neutron activation analysis [6,7], and inductively coupled plasma mass spectrometry (ICP-MS) [3,8]. Laser ablation inductively coupled plasma mass spectrometry [7,9] and micro X-ray fluorescence spectrometry (µXRF) [10] can be used for separate analysis of the clay component and inclusions in the ceramic cross-section. Non-destructive characterization of bulk ceramic composition is possible using portable XRF [11,12,13,14,15].

Despite the large number of methods used to analyze ceramics, one of the most popular approaches to chemical analysis in archaeology is the use of XRF spectrometry combined with machine learning techniques to attribute samples to specific places of origin. This popularity is due to the relative ease and accessibility of XRF analysis, its non-destructive nature, its ability to be performed in the field, and the wide range of elements determined. Machine learning methods (chemometric methods, in the context of chemical analysis) allow the extraction of complex and non-obvious correlations between objects and their characteristics in the large volume of data provided by XRF [16]. The arsenal of modern chemometric methods is quite large and includes methods of clustering, classification, and multivariate regression. The choice of specific methods of investigation depends on the problem to be solved. In recent years, the combination of XRF and chemometric methods has become increasingly common in archaeological research. Typical recent works are worth mentioning here. In [17], artificial neural networks (ANNs) and linear discriminant analysis (LDA) methods were successfully applied in combination with XRF and XRD data to attribute the origin of pottery samples to different Greek colonies in Sicily. The accuracy of the classification using ANN was 78% and that of LDA was only 50%. In [18], the potential of two different instrumental methods for hierarchical cluster analysis (HCA) of Medieval and post-Medieval ceramics from the Iberian Peninsula was compared. It was shown that the simpler, non-destructive XRF method achieves almost the same clustering efficiency as the more complex, expensive, and destructive ICP-MS method.

The most popular XRF instruments for archaeological tasks are energy dispersion instruments [18,19,20,21,22,23], including portable XRF instruments [11,12,13,14,15]. These devices offer reliable determination of major elements, but do not provide information about the contents of impurities and trace elements, which, in the context of archaeological research, can be more informative. This problem can be solved by using total-reflection X-ray fluorescence spectrometry (TXRF), the detection limits of which are several orders of magnitude lower than those of traditional XRF. TXRF has been successfully applied to determine the chemical composition of ceramics [24,25,26,27,28]. In these works, it was shown that the limit of detection and limit of quantification of TXRF are adequate for the determination of trace elements in these samples.

Recently, studies have explored the combination of TXRF with chemometric methods as a possible tool for the analysis of complex samples. This combination was already successfully applied in the discovery of obesity markers [29], identification of the geographical origin of bean seeds [30], classification of the origin and type of wine samples [31], discrimination of gunshot residues [32], and seafood provenance assessment [33,34]. This approach was also applied to the analysis of archaeological ceramics [35], but the data in this work were investigated using principal component analysis (PCA), which is an exploratory method and does not allow the classification of samples. PCA can only be considered an unsupervised clustering technique, as it does not require the knowledge of class attribution in the data under analysis [36].

The goal of the present study was to further elaborate the combination of TXRF and chemometrics and to check whether this approach is suitable for fast and reliable provenance analysis of archaeological ceramic samples.

## 2. Results and Discussions

### 2.1. TXRF Analysis

This research was performed using a TXRF method for the analysis of ceramic samples, successful validation of which was performed in our previous study [35]. The analytical features of the method were not under consideration in this study. Table 1 presents the concentration ranges of the elements in the 81 ceramic samples obtained by TXRF analysis and divided by archaeological sites. Most of the ceramic samples (45) are represented by those from the Popovsky Lug site. Elements of interest were determined not only by the sensitivity and accuracy of the TXRF method, but also by the previous investigations, where element indicators were found [35]. The acid leaching sample preparation of the ceramic allows transferring of the clay component into solution and separation of the insoluble silicate minerals included in the ceramic [24,35]. Using this approach for provenance analysis allows for better identification of sample groups than using the bulk composition of ceramic samples. This method, based on the analysis of the clay components of ceramics, is a perspective to be applied for comparison with the chemical composition of regional clays.

Figure 1 shows the spectra of ceramic samples from different sites. To perform the correct interpretation of spectra, all intensities were normalized to the Se-Kα line (internal standard). As can be seen from the spectra, there is either a slight (Ni, Ga, Y, and Pb) or great (Cr, Zn, and Sr) difference among peaks of same elements for samples from different sites. This means that ceramic samples may have similar elemental compositions even if they were found far away from each other.

### 2.2. Factor Analysis and Principal Component Analysis (PCA)

Factor analysis and PCA chemometric techniques are among the most popular techniques for exploratory data analysis. PCA is employed as a data dimensionality reduction method to visualize the samples in a lower dimension space. The data obtained by TXRF for the 81 ceramic samples were composed in a matrix that contained the concentrations of 10 elements as well as the characteristics of archaeological types and sites. First, factor analysis was applied for fast pattern recognition. It allowed the extraction of four significant factors, and showed that three elements, according to the first three factors, had the highest factor loadings: Zn, Cr, and Sr. Ternary diagrams could be constructed using these three factors. Figure 2 presents two ternary diagrams using Zn, Cr, and Sr concentrations divided by archaeological site and type.

Looking at Figure 2a, one may recognize a cluster of samples from the Ust-Karenga site in the left-bottom corner. The Popovsky Lug site covers most of the space in Figure 2a and overlaps with other sites. However, one may visually separate the Shishkino and Makarovo sites from the others. If we look at the right ternary diagram (Figure 2b), which was made by type of ceramics, we may recognize the same cluster of the Ust-Karenga type. There was no more distinct clustering observed for the other types of samples.

Next, we applied principal component analysis (PCA). The study of the score and loading scatter plots allowed the identification of criteria for clustering objects. Figure 3a–d present the score scatter plots that were constructed by projections on the first and second principal components of PC1–PC2. Figure 3e–h present the corresponding loading plots. Figure 3a,c are plotted by site; Figure 3b,d are plotted by type. The cumulative percent of variance in correlation matrix for the first two principal components is about, or more than, 50% for all scatter plots. In Figure 3e,f, the most significant variables for the first PC are Cr, Ni, V, and Pb. In Figure 3g,h, the most significant variables for the first PC are V, Rb, and Pb. The second principal component divides the samples by the contents of Ni and Sr.

As can be seen from Figure 3a, ceramics from the Popovsky Lug, Shishkino, Ust-Yamniy, and Makarovo sites are practically the same, while the ceramics from Ust-Karenga and Ust-Yumurchen are separated from them. For Ust-Karenga ceramics, the high content of Pb in the samples is noteworthy. The samples from Ust-Karenga differ from the others in the content of the groups of elements correlating with each other: Rb, Pb, Ni, V, and Cr. The samples from the Shishkino, Ust-Yamniy, and Makarovo sites form a common cluster with those from the Popovsky Lug site. These sites, located in the uppermost (southern) section of the upper Lena, are fairly close to each other, and this fact can explain the similarity in the PCA graphs. If we exclude the samples from Popovsky Lug, we may discover a clear segregation of the samples from Makarovo, Shishkino, and Ust-Karenga (Figure 3c). 

The division of ceramic samples by archaeological type is shown in Figure 3b (by all types) and in Figure 3d (only samples from Popovsky Lug). The scatter plot in Figure 3b shows a clear differentiation of the Ust-Karenga ceramic type. Figure 3d shows that the Setchaty type is a little apart from the other samples. The Ust-Belsky and Posolskaya types are distributed over the entire area of the graph, which makes it impossible to judge the difference between these samples. The rest of the samples are mixed.

Despite some useful results being found, factor analysis and PCA did not show any systematic pattern able to create reference groups according to common provenance features.

### 2.3. Generalized Cluster Analysis by k-Means (k-Means CA)

The application of k-means CA is a fast and easy way for analyzing provenance without preliminary preparation of a dataset, for example, excluding outliers. In the case of PCA, we found similarities in the ceramic samples from different sites geographically close to each other without the possibility of diverse results. For example, ceramic samples from Popovsky Lug, Shishkino, and Makarovo could not be separated. The k-means CA gave us a better view on the differences in the ceramic samples.

Table 2 presents the results of the k-means CA of ceramic samples, characterized by archaeological site. Testing error was equal to 0.448; training error—0.272. The total number of clusters was six, which is the same number as the number of archaeological sites. The Popovsky Lug site and Makarovo site are only related to cluster Nos. 3 and 4, respectively. These two sites are clearly distinguished from the others, which was not possible using PCA. Most of the Ust-Karenga site samples (7/8) are related to cluster No. 1. Samples from the Ust-Yamniy site share cluster No. 6 with the samples from Shishkino, as was observed by PCA. Two of the four Ust-Yumurchen samples are in a cluster No. 5, and one of the four belongs to cluster No. 6. Cluster No. 2 only includes two samples from Ust-Karenga and Ust-Yumurchen, the same pattern observed on the PCA graphs. In total, the results of k-means CA are better and clearer for provenance analysis than those of PCA. The method of k-means CA also enables the elimination of the subjective factor in the assessment of classification and the following additional errors.

Table 3 presents the results of the k-means CA of ceramic samples, characterized by type. The conditions of analysis were the same as in the case of analysis by site. Testing error was equal to 0.349; training error—0.206. The total number of clusters was five. The results of the classification of ceramic by type were even better than those by site. Clear and obvious clusters were obtained for the Ust-Belsky, Setchaty, Khaitinsky, and Posolskaya sample types. Only the Ust-Karenga type is divided by two clusters, but the clusters are independent. An independent cluster means no correlation with other variables.

The chemometric technique of k-means CA works very well with ceramic samples based on elemental data. The classification of ceramics by type is better than by site, meaning that archaeological interpretation based on visual assessment of found material is reliable.

### 2.4. Support Vector Machines (SVMs)

Based on the results of SVM, Table 4 and Table 5 present a differentiation of the ceramic samples by archaeological site and type, respectively. Some samples were not correctly classified, and the tables also contain the percent of misclassification and incorrect attribution.

Only four outliers were detected by the PCA method for the SVM model of samples, characterized by site. The classification accuracy of SVM performed for sites is 84.416%. In the summary table of the SVM method, some samples from the sites of Ust-Yamniy and Makarovo are defined as Popovsky Lug and vice versa, although they have some of their own characteristics associated with the type of pottery. The pottery of the Ust-Karenga site stands out in terms of chemical composition, although the pottery of the Ust-Yumurchen site is not clearly distinguished. In conclusion, data were obtained for four samples from Ust-Yumurchen, which was not enough for unambiguous conclusions. 

Table 5 presents the results of the SVM analysis of the ceramic samples by archaeological type. Six outliers were excluded before construction of the SVM model. The classification accuracy of the SVM as performed as 89.333%. The Khaitinsky, Posolskaya, and Ust-Karenga types were clearly distinguished by SVM. In total, 2 of the 13 samples of the Ust-Belsky type were incorrectly classified as being of the Posolskaya type. Almost half of the Setchaty-type samples were wrongly classified as the Khaitinsky or Posolskaya type.

The SVM model worked very well for the provenance analysis of ceramic samples, with an accuracy of more than 80%. To find out why the incorrectly predicted samples did not fit into the classification, we repeated the PCA after removing outliers. Those ceramic samples were indeed located at the edges of the 3σ border. If we removed the ceramic samples incorrectly predicted by the SVM method, then almost 100% accuracy of the SVM method was obtained. Thus, the resulting database using the SVM method can be supplemented with new samples for subsequent automatic classification.

## 3. Materials and Methods

### 3.1. Sample Description

A total of 81 samples of archaeological ceramics were taken from sites located in the valleys of the Upper Lena River and Upper Vitim River. The geographical locations of the sites are presented on a map (Figure 4). The multilayer archaeological sites of Popovsky Lug, Makarovo, Shishkino, and Ust-Yamniy represent the main types of ceramics (Setchaty, Khaitinsky, Ust-Belsky, and Posolskaya) located in the part of the Upper Lena in the Kachugsky district of the Irkutsk region. Ceramic samples at these sites are represented by the main types of Neolithic ceramics of the Baikal region. The Setchaty type is characterized by oval and parabolic vessels of closed or open form, with an imprint of wicker mesh on the outer surface of the vessel. The Khaitinsky type is characterized by a complex shape with a cambering of the upper part and a pointed bottom, with the outer surface covered by imprints of a thin twisted cord, formed as a result of knocking out with a wrapped mallet and decorated by drawing and pricking. The ceramics of the Posolskaya type are characterized by a cambering of the upper part of the vessels and have a pointed or rounded bottom; on the outer surface, there are traces of a ribbed blade or a blade wrapped with a cord. A distinctive feature of Posolskaya-type ceramics is a subtriangular thickening on the outer or inner side of the rim, under the cut of which a belt of through-punctures is applied along a drawn or pressed groove. Ceramics of the Ust-Belsky type are characterized by both closed- and open-shape vessels with pointed or rounded bottoms. The vessels are smooth-walled, decorated from the rim to the bottom, and distinguished by horizontal and zigzag rows of imprints made with a stack.

Ceramics from the Ust-Karenga and Ust-Yumurchen sites are represented by corded pottery. These two sites are located near the Upper Vitim River within the Vitim Plateau (north-eastern Transbaikalia). Ust-Karenga ceramics have a paraboloid shape with imprints of a thin twisted cord on the outer surface and traces of rubbing on the inner side. The ornament is made with a comb stamp; a zigzag shape is dominant. Ust-Yumurchen ceramics are characterized by imprints of a ribbed blade or a twisted cord on the outer surface. The ornament was applied with a jagged stamp as well as a rectangular or oval stack in cross-section. The rim has a subtriangular molding on the outer or inner side, which makes it similar to Posolskaya-type ceramics.

The selection of samples was carried out mainly with reference to the vessels for a full description and archaeological interpretation. The ceramic fragments were taken from different parts of the vessels and were rinsed in de-ionized (18.2 MΩ) water in an ultrasonic bath for 60 min, dried, and photo-documented. Typical ceramic samples from the aforementioned archaeological sites are presented in Figure 5. Then, the ceramic samples were crushed and milled using a mortar grinder.

### 3.2. Sample Preparation

The following reagents were used for TXRF analysis: single-element standard solution of Se (C = 1000 mg/L, Merck, Darmstadt, Germany) for the preparation of the internal standard; nitric acid and hydrochloric acid (ultra-pure grade, Merck) for the preparation of the aqua regia used in the leaching procedure; and ultrapure deionized water (18.2 MΩ, Elga Labwater, High Wycombe, UK) for dilution.

We placed 20 mg of the sample in a polytetrafluoroethylene (PTFE) vessel, and 1 mL of aqua regia was added. The closed vessel was heated on a plate for 8 h at a temperature of 170 °C, cooled, and then 3.95 mL of ultrapure water and 0.05 mL of the internal standard of Se were added. Homogenization of the obtained solution was performed using a Vortex (IKA, Staufen im Breisgau Germany) for 5 min. The sediment in the form of insoluble solids was separated. An amount of 10 µL of the solution was put onto a quartz carrier and dried on a heating plate.

### 3.3. Spectra Acquisition

TXRF elemental analysis was performed using a benchtop spectrometer S2 PICOFOX (Bruker Nano Analytics, Berlin, Germany) equipped with an X-ray tube with a Mo-anode, multilayer monochromator, and 30 mm^2^ silicon drift detector (energy resolution was <150 eV at Mn-Kα line). Measurement of one sample was carried out in triplicate during 500 s at a 50 kV voltage and a 0.50 mA current. Quartz carriers were used as sample holders and reflectors. The Spectra 7.8.2 software package (Bruker Nano Analytics, Berlin, Germany) was applied for spectra processing.

### 3.4. Data Analysis

The concentration of the element (*C_i_*, mg/kg) was calculated using the following formula [37]:(1)$$C_{i}=\frac{C_{is}\cdot N_{i}\cdot S_{is}}{N_{is}\cdot S_{i}}$$
where *C_is_* is the concentration of the internal standard; *N_i_* and *N_is_* are the net peak areas of an element of interest and the internal standard, respectively; *S_i_* and *S_is_* are the sensitivities of an element of interest and internal standard, respectively.

The following elements were excluded from the data due to the study [38] of heterogeneity of ceramic samples: Ca, P, and Mn. In [38], it was shown that concentrations of these elements may strongly vary within the same sample and, therefore, cannot be considered in a provenance analysis.

The database of ceramic samples, with a matrix of 81 × 11, was divided into separate files, each representing a specific type/site of pottery, and then given statistical treatment. Different classification approaches were used for provenance analysis: unsupervised learning and pattern recognition (factor analysis, PCA) and supervised learning (k-means cluster analysis and support vector machines). Unsupervised learning algorithms are not guided by previously known classifications; only after certain clusters have been defined is it possible to assign labels. Supervised methods fit the model to a given classification. Chemometric methods were performed using the STATISTICA 10 program (TIBCO Software Inc., Palo Alto, USA).

Factor analysis (FA) was applied under the following conditions: 10 variables (elements), extraction of 3 factors, method of principal components, and varimax raw factor rotation.

Principal component analysis (PCA) was applied as follows: 10 elements were chosen as variables, scores were automatically standardized, and loading and score plots were based on the first two components.

The k-means cluster analysis (k-means CA) was applied under the following conditions: k-means algorithm, squared Euclidian distance method, and training/test samples 2:1.

Support vector machines (SVMs): outliers were determined using the PCA method. Figure 6 shows an example of outlier detection by the construction of score and loadings plots in the space of principal components. Samples outside of the 3σ border were considered outliers. After exclusion of all the outliers, the SVM model could be applied.

The SVM method is based on data mining and machine learning. By default, the program divides the entire set into 2 groups: training and testing. We placed 75% of the samples into the training group, and the remaining 25% into the test group. The SVM model characteristics were classification type 2, the kernel type was a radial basis function (gamma = 0.450), and cross-validation.

## 4. Conclusions

A reference database was created to characterize the ancient ceramics of the studied territories. The database was integrated into the STATISTICA program for the rapid study of the differences and similarities of the samples using chemometric approaches (PCA, k-means CA, and SVM), as well as for classification by geographical origin and ornamentation. The study showed that unsupervised clustering methods such as PCA can be used for an initial assessment of a matrix, while k-means CA or SVM should be applied as primary methods for provenance analysis. According to the results of these chemometric techniques, some samples from the Popovsky Lug, Ust-Yamniy, and Makarovo sites formed a common cluster, because the sites are very close to each other and have the same clay sources. These sites are in the uppermost (southern) section of the upper Lena, which can be conditionally called Kachugsky after the regional center of the village of Kachuga. The fairly close location of the objects to each other explains the similarity in geomorphological position, the correlation of stratigraphic sections, and the similar composition of clay taken from different objects of this region. The chemical composition of pottery from the Ust-Karenga and Ust-Yumurchen sites (some of the samples) differs from that of pottery from other sites due to the differences in ownership and sources of raw materials. For a more detailed study of ceramics, it will be necessary to analyze a larger number of samples from this site, as well as clay sources. Based on the results obtained in this study, we may assume that the raw materials used for the manufacture of the ceramic samples were local. The database created and processed by the SVM or k-means CA methods can be supplemented with new samples and then automatically classified. This study showed that a combination of the TXRF method and chemometric techniques is a fast and effective way to conduct provenance analysis of archaeological ceramics. 

## Figures and Tables

**Figure 1 molecules-28-01099-f001:**
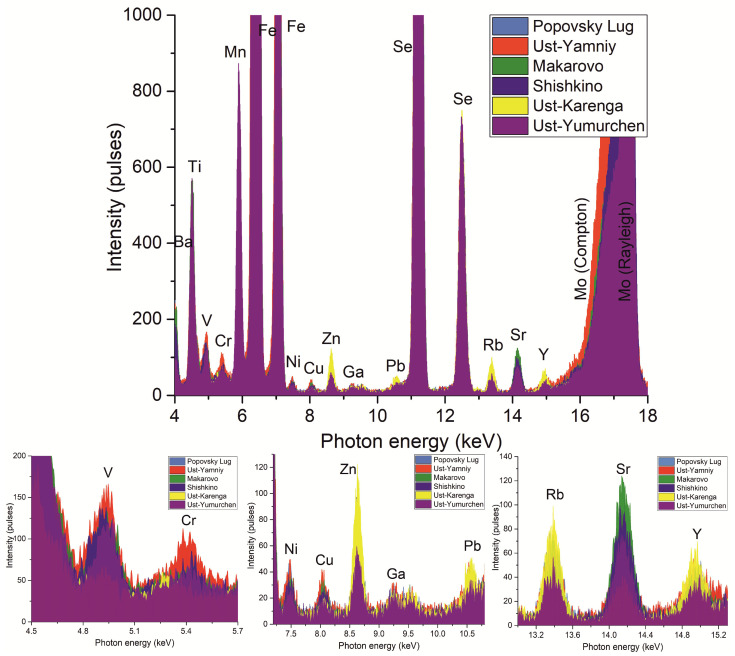
TXRF spectra (energy range: 4.0–18 keV) of ceramic samples, one of each from different archaeological sites.

**Figure 2 molecules-28-01099-f002:**
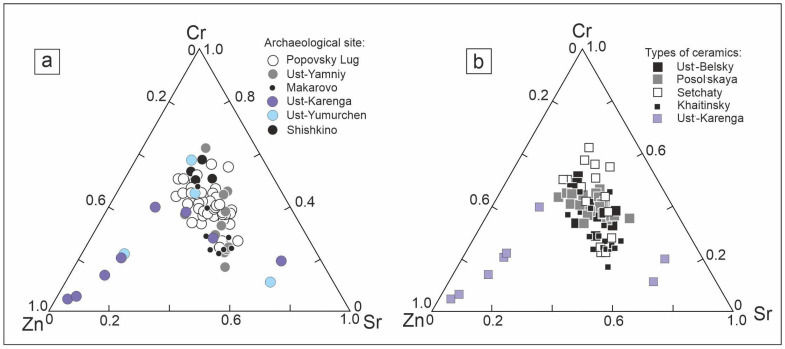
Ternary diagrams for Zn, Cr, and Sr divided by archaeological site (**a**) and type (**b**).

**Figure 3 molecules-28-01099-f003:**
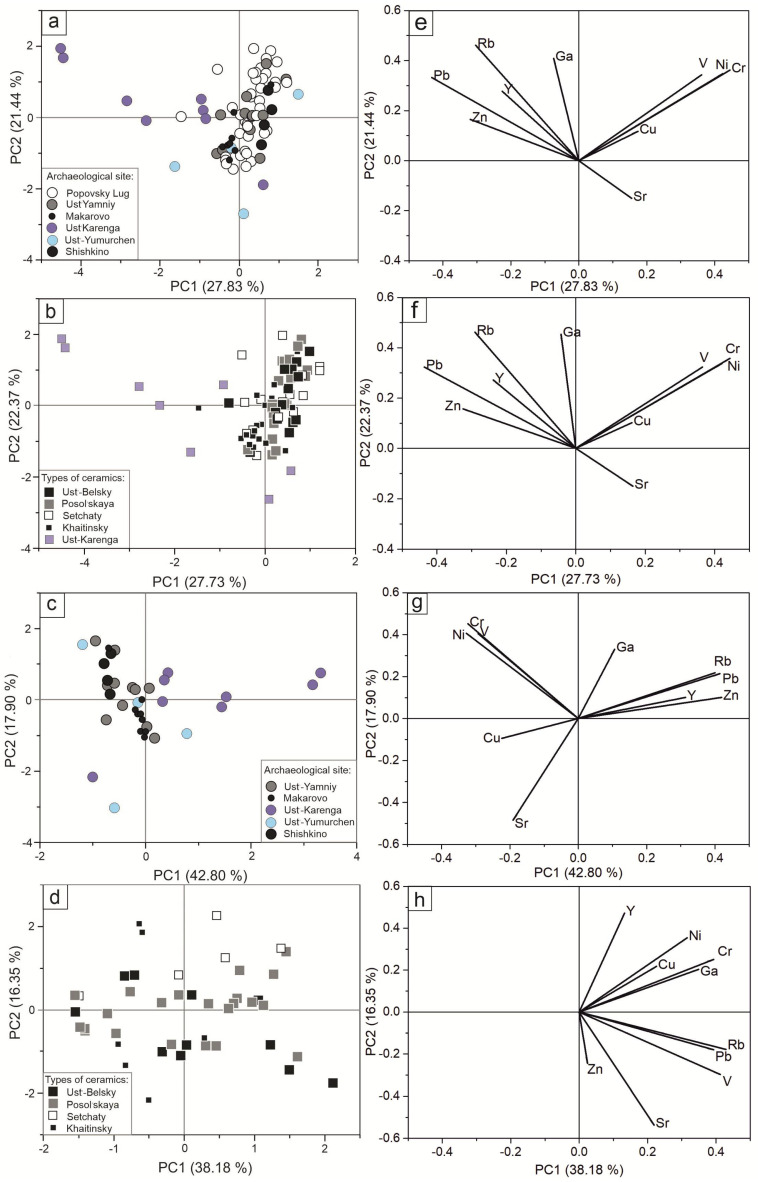
PCA score (**a**–**d**) and loading (**e**–**h**) plots calculated for elemental composition of ceramic samples performed with division by archaeological site ((**a**,**e**)—all samples; (**c**,**g**)—samples without Popovsky Lug) and type ((**b**,**f**)—all types; (**d**,**h**)—samples from Popovsky Lug).

**Figure 4 molecules-28-01099-f004:**
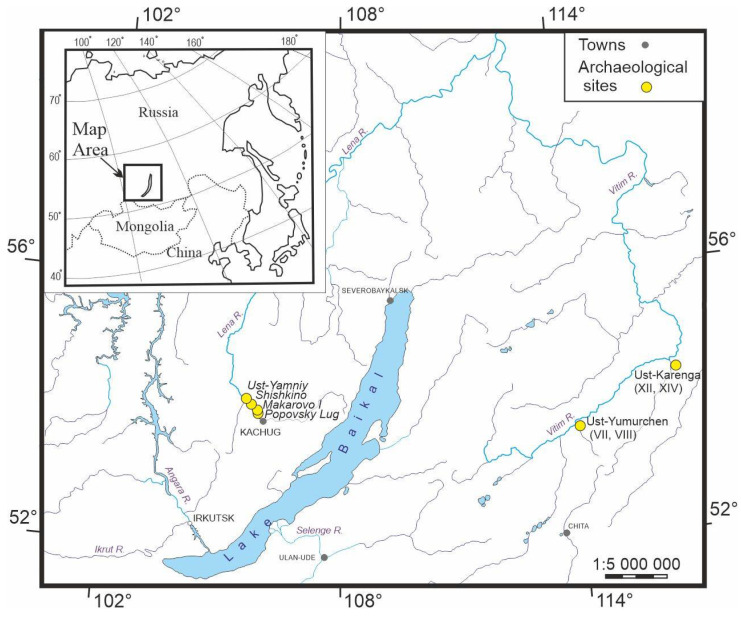
Map of archaeological sites.

**Figure 5 molecules-28-01099-f005:**
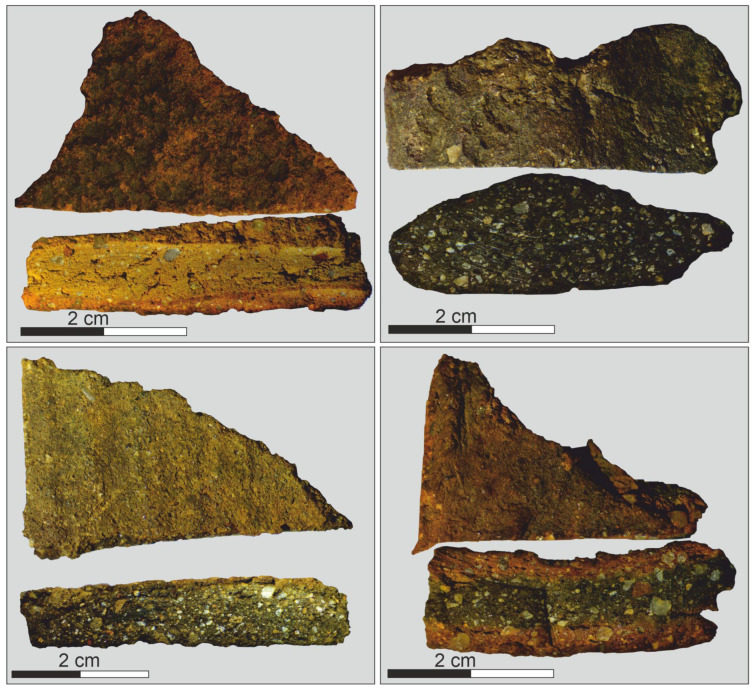
Examples of pottery sherds from eastern Siberia (Baikal region).

**Figure 6 molecules-28-01099-f006:**
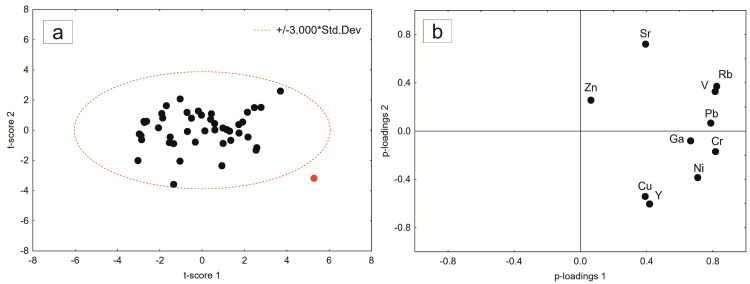
Score (t1–t2) (**a**) and loading (p1–p2) (**b**) plots in the space of principal components.

**Table 1 molecules-28-01099-t001:** Elemental characterization of ceramic samples by TXRF method.

Archaeological Site (Number of Samples)	Concentration Range (mg/kg)									
	V	Cr	Ni	Cu	Zn	Ga	Rb	Sr	Y	Pb
Popovsky Lug (45)	31–152	57–172	9–82	9–62	26–943	8–19	13–76	36–203	7–57	19–72
Ust-Yamniy (11)	40–162	50–193	15–50	12–50	37–129	4–18	13–60	64–188	14–30	26–66
Makarovo (9)	39–126	62–142	16–44	12–31	58–114	10–15	24–63	69–168	13–27	27–51
Ust-Karenga (8)	0–63	26–98	6–30	12–61	38–501	11–23	20–122	21–201	3–52	21–180
Ust-Yumurchen (4)	16–99	28–163	5–59	12–42	62–83	8–14	10–43	18–260	9–20	18–49
Shishkino (4)	54–97	90–138	30–48	21–31	46–67	10–15	23–48	42–76	8–21	25–42

**Table 2 molecules-28-01099-t002:** Frequency table for cluster analysis of ceramic samples characterized by archaeological site.

	Cluster Number	1	2	3	4	5	6	Total
Archaeological Site								
Popovsky Lug		0	0	45	0	0	0	45
Ust-Yamniy		0	0	0	0	0	11	11
Makarovo		0	0	0	9	0	0	9
Ust-Karenga		7	1	0	0	0	0	8
Ust-Yumurchen		0	1	0	0	2	1	4
Shishkino		0	0	1	0	0	3	4

**Table 3 molecules-28-01099-t003:** Frequency table for cluster analysis of ceramic samples characterized by type.

	Cluster Number	1	2	3	4	5	6	Total
Type								
Ust-Belsky		14	0	0	0	0	0	14
Posolskaya		0	0	0	26	0	0	26
Setchaty		0	0	0	0	15	0	15
Khaitinsky		0	18	0	0	0	0	18
Ust-Karenga		0	0	6	0	0	2	8

**Table 4 molecules-28-01099-t004:** Differentiation of ceramic samples by archaeological site based on the results of SVM.

Archaeological Site	Correct Attribution, Total	Correct Attribution %	Misclassification %	Incorrect Attribution Site
Popovsky Lug	39/41	95%	5%	Makarovo/Ust-Yamniy
Ust-Yamniy	7/11	64%	36%	Makarovo/Popovsky Lug
Makarovo	7/9	78%	22%	Popovsky Lug
Ust-Karenga	7/8	88%	12%	Ust-Yumurchen
Ust-Yumurchen	1/4	25%	75%	Ust-Karenga/Popovsky Lug
Shishkino	4/4	100%	0%	-

**Table 5 molecules-28-01099-t005:** Differentiation of ceramic samples by archaeological type based on the results of SVM.

Archaeological Type	Correct Attribution, Total	Correct Attribution %	Misclassification %	Incorrect Attribution Type
Khaitinsky	15/15	100%	0%	-
Setchaty	9/15	60%	40%	Khaitinsky /Posolskaya
Ust-Belsky	11/13	85%	15%	Posolskaya
Posolskaya	24/24	100%	0%	-
Ust-Karenga	8/8	100%	0%	-

## Data Availability

The data obtained are available via the author contacts.

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
