# Peer review of "Combination of Total-Reflection X-Ray Fluorescence Method and Chemometric Techniques for Provenance Study of Archaeological Ceramics"

_molecules, 2023, doi:10.3390/molecules28031099_

Round 1
Reviewer 1 Report
The article reports the applicability and performance of total-reflection X-ray fluorescence combined with chemometrics (e.g., FA, PCA, generalized CA, SVM) to investigate Neolithic ceramic samples from eastern Siberia. The development of fast and non-destructive methods for the authentication and characterization of archeological samples of different origin can be surely of interest both for historical and civilization research.
The “Introduction” section basically provides a good summary of the relevant literature and the relevance of the topic However, some questions arise, especially regarding the “Materials and Methods” section. After the necessary additions and corrections mentioned below are made, the article can be reconsidered for publication. Please find below my remarks and questions on the study.
Lines 72-77: When formulating objectives, it is not necessary to list the materials and methods used, since it is needed to be detailed in the “Materials and Methods” section.
2. Materials and Methods: Reformulate this section similarly to this: 2.1 Sample description, 2.2. Sample preparation, 2.3. Spectra acquisition, 2.4. Data analysis. First, describe what samples were tested, where they came from and how they were prepared for the measurements. You mention that there are photos of the samples tested, it might be worth including some. Then go into much more detail about how the samples were measured, how they were placed into the instrument, whether and how many times the measurements were repeated, etc. In a subsequent subsection, detail what statistical analysis was used, how the data were processed, which chemometric methods were used for what purpose, and how the models were constructed and validated.
Combine the information included in section 2.2. and 2.4.
Please include the data pre-processing steps mentioned in section 3.1. (exclusion of certain elements based on former research findings) in “Materials and Methods”.
Figure 3.: Why did not you apply 3D visualization of the results?
Section 3.2.: The format of notation a-d in the figures referred to in the text are different, please unify.
Line 225: Write out what the CA abbreviates (e.g., cluster analysis (CA)).
Section 3.4.: Include the data pretreatment steps prior to actual SVM in the relating part of Materials and Methods (e.g., 2.4. Data analysis).
Figure 5.: Include a and b letters in the figure and caption.
Lines 286-287: Phrase “For a more detailed study of ceramics, it is necessary to analyze a larger number of samples from this site, as well as clay sources.” rather belongs to Conclusions.
It is possible to know from the SVM results which compounds were the most relevant in the analysis. It would be worth listing them, as they were mentioned in the FA and PCA analyses.
4. Conclusions: Please add your experience of which statistical methods (e.g., CA and/or SVM) have proved most successful in your study.
It is strongly recommended that in the (near) future, completely non-destructive (spectroscopy) and even non-contact methods (hyperspectral imaging) could be used to avoid damaging archeological samples.
Once the necessary additions, clarifications and amendments have been made, the manuscript can be considered for publication.
Reviewer 2 Report
My comment is attached herewith in a word file.
